# Entomo-virological investigation during the epizootic outbreak of sylvatic yellow fever in Rio Grande do Sul, Brazil, between 2021 and 2022

Nícolas Felipe Drumm Müller[1,2,/+], Marcelo de Moura Lima[2], Edmilson dos Santos[2], Aline Alves Scarpellini Campos[2], Thomas Rosa Menegazzi[2], Alanis Silva Melgarejo[1], Bruna Paredes-Galarza[1], Lina Violet-Lozano[1], Martha Trindade Oliveira[1], Cirilo Henrique Oliveira[3], Paulo Michel Roehe[1†], Fabrício Souza Campos[1], Filipe Vieira Santos de Abreu[3,4], Jáder da Cruz Cardoso[2], Ana Cláudia Franco[1/+]

[1]Universidade Federal do Rio Grande do Sul, Instituto de Ciências Básicas da Saúde, Laboratório de Virologia, Porto Alegre, RS, Brasil
[2]Secretaria Estadual de Saúde do Rio Grande do Sul, Centro Estadual de Vigilância em Saúde, Porto Alegre, RS, Brasil
[3]Instituto Federal do Norte de Minas Gerais, Laboratório de Comportamento de Insetos, Salinas, MG, Brasil
[4]Fundação Oswaldo Cruz-Fiocruz, Instituto Oswaldo Cruz, Laboratório de Mosquitos Transmissores de Hematozoários, Rio de Janeiro, RJ, Brasil

**BACKGROUND** Yellow fever virus (YFV) re-emerged among non-human primates (NHPs) in Rio Grande do Sul in early 2021, more than a decade after its last detection in the state. The spread of the virus was accompanied by increased mortality among NHPs.

**OBJECTIVES** To conduct entomological surveillance and molecular detection of YFV and other *Orthoflavivirus* species in mosquito samples collected from affected and potentially receptive areas.

**METHODS** Mosquitoes were collected during epizootics using human landing catches, BG-Pro traps, and ovitraps. Virus detection was performed using reverse transcription real-time polymerase chain reaction (RT-qPCR) assays targeting YFV and pan-*Orthoflavivirus* sequences.

**FINDINGS** A total of 1,210 mosquitoes, representing 26 taxa, were collected across 17 municipalities. *Psorophora ferox* was the most abundant species, followed by *Culex* (*Culex*) spp., accounting for 27% and 12% of the specimens, respectively. *Haemagogus leucocelaenus*, the primary YFV vector in the region, was also among the most frequently captured species, representing 7%. In total, 203 mosquito pools were assembled by species, location, and date of collection. RT-qPCR analysis did not detect YFV or other *Orthoflavivirus* RNA in any of the samples.

**MAIN CONCLUSIONS** Although mosquitoes were collected during a period of active YFV circulation, the absence of virus detection suggests that arboviral circulation in vector populations may occur at low frequencies, even during outbreaks.

Key words: yellow fever virus - arboviruses - Culicidae - mosquito vectors - viral zoonosis

*Orthoflavivirus flavi*, widely known as yellow fever virus (YFV), is the prototype virus of the genus *Orthoflavivirus*, family *Flaviviridae*. It is an endemic arbovirus in tropical and subtropical regions.[1] In South America, YFV is maintained through two distinct transmission cycles: urban and sylvatic. The urban cycle involves the mosquito *Aedes aegypti* and humans.[2] Due to the temporary eradication of *Ae. aegypti* between the 1940s and 1960s, along with human vaccination efforts, the urban transmission cycle was interrupted, and the virus remained confined to forested areas, where the sylvatic cycle persists. In these areas, native mosquitoes of the genera *Haemagogus* and *Sabethes* serve as vectors, non-human primates (NHPs) act as amplifier hosts.[3] In Brazil, NHPs are used as sentinels for YFV surveillance due to their high susceptibility to infection.[4] Humans, within the sylvatic cycle, are considered incidental hosts, with infections occurring sporadically, developing disease and viraemia, although they generally do not sustain transmission.[5]

In Brazil, YFV is endemic in the Amazon rainforest, but epidemic activity has extended to the southern limits of the Atlantic Forest. YFV circulation in extra-Amazonian regions has been reported since the early 2000s.[3] From 2014 onwards, the virus expanded from the Cer-

Financial support: The project was supported by grants from CNPq/Departamento de Ciência e Tecnologia/Secretaria de Ciência, Tecnologia e Insumos Estratégicos/Brazilian Ministry of Health (CNPq/Decit/SCTIE/MS) (grant number 443215/2019-7 to ACF, PMR and FSC).
† *In memoriam*
+ Corresponding authors: nicolas-muller@saude.rs.gov.br | ⬡ https://orcid.org/0000-0002-6322-252X
anafranco.ufrgs@gmail.com | ⬡ https://orcid.org/0000-0001-7747-5125

**Handling editor:** Ademir de Jesus Martins Jr | ⬡ https://orcid.org/0000-0001-5739-1215

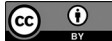

rado biome in northern Brazil to other regions, causing high mortality in NHPs.[3] Between 2016 and 2019, YFV reached the Atlantic Forest in the southeast, the most densely populated region of the country, triggering the largest sylvatic YFV outbreak ever recorded in Brazil.[4,6] During this period, more than 2,000 human cases and over 700 fatalities were reported, this significant outbreak was primarily attributed to the lack of vaccination among the local population.[7] Since 2019, YFV has also been detected among NHPs in southern Brazil[8] and, in early 2021, the virus reached Rio Grande do Sul (RS), Brazil's southernmost state, after more than a decade without recorded activity.[9]

In the 21st century, the first YFV epizootic outbreak in RS occurred between 2001 and 2002, and was restricted to the northwestern region of the state. During this event, NHP deaths were recorded, but no human cases were reported.[10] The largest sylvatic YFV outbreak in the RS took place between 2008 and 2009. Again, NHP deaths began in the northwest and subsequently spread to central and northern regions.[11] This outbreak resulted in 21 confirmed human cases, including nine fatalities. Additionally, 204 NHP deaths were confirmed by laboratory diagnosis, although the number of NHPs affected is estimated to exceed 2,000.[12] In the 2021 epizootic outbreak, the YFV dispersion wave reached the northeastern region of RS via neighbouring states, a marked shift from previous outbreaks, which typically entered through the northwest, bordering Argentina.[9] The virus then progressed southward, and also affected areas in the central and northwestern regions of the state.[13] In total, 420 NHP deaths were reported in 2021, of which 126 were laboratory-confirmed as YFV-positive. Notably, the majority of NHP cases occurred during the first semester of the year, with a marked decline in reports coinciding with the onset of the colder season, which is unfavourable for mosquito activity.[13] As a result of prior vaccination efforts and heightened public awareness, no human infections were reported throughout 2021.[13]

In addition to the circulation of YFV, there is evidence supporting the presence of other orthoflaviviruses in RS. In urban settings, dengue virus, transmitted by *Ae. aegypti*, has been circulating autochthonously in humans since 2007, with a marked increase in human incidence observed from 2021 onward.[13,14] Zika virus has also been generating human cases since 2017, although with a lower incidence compared to dengue.[13,15] Although no human cases have been officially reported in RS for other orthoflaviviruses, indirect historical evidence suggests their circulation among wild and livestock animals, indicating potential zoonotic risk. Antibodies against *Orthoflavivirus nilense* (West Nile virus, WNV) have been detected in birds and horses,[16,17] and antibodies for *Orthoflavivirus louisense* (St. Louis encephalitis virus, SLEV) and *Orthoflavivirus ilheusense* (Ilheus virus, ILHV) have been found in NHPs.[18,19] Considering the evidence of arbovirus circulation in RS, this study aimed to analyse sylvatic mosquitoes collected during the YFV outbreak for the presence of YFV and other orthoflaviviruses.

## MATERIALS AND METHODS

*Study area* - Rio Grande do Sul is the southernmost state in Brazil, bordering Uruguay and Argentina. The state covers an area of 281,707 km² and comprises 497 municipalities, with an estimated population of over 10 million people. Two major biomes occur in RS: the Pampa, a grassland biome where forests are limited to riparian zones and isolated patches, and the Atlantic Forest, a forest biome influenced by its proximity to the ocean.[20]

*Mosquito sampling and taxonomic identification* - The study was conducted with support from the Environmental Health Surveillance Division (DVAS) of the Rio Grande do Sul State Health Surveillance Centre (CEVS). Mosquito collections were carried out from February 2021 to February 2022 during the summer and spring seasons in forested areas of several municipalities in RS. Due to the emergency context of an active YFV outbreak, sampling procedures were not standardized across locations. Field strategies were dynamically adjusted based on real-time notifications of sick or dead NHPs recorded by DVAS, prioritizing newly affected areas (Fig. 1). Mosquito sampling was carried out concurrently with epizootic investigations and up to 25 days thereafter. Additionally, collections were extended to ecologically connected municipalities, areas with YFV circulation in previous years and two municipalities [Pinhal da Serra (PDS) and Esmeralda (ESM)] revisited approximately one year after the initial epizootic events [Supplementary data (Table)].

In all areas, adult mosquitoes were collected using the human landing catch (HLC) method, which relies on the natural production of carbon dioxide ($CO_2$), heat, and human odour to attract mosquitoes seeking hosts. Between two and five collectors performed the HLC method using entomological nets and manual aspirators, operating during the day (9:00 a.m.-4:00 p.m.), for variable periods depending on field conditions.

At municipalities Santo Antônio das Missões (SAM), Derrubadas (DER), and ESM, additional collections were conducted using BG-Pro traps (Biogents AG, Regensburg, Germany). Each trap was baited with 2 kg of dry ice (as a $CO_2$ source) and an artificial attractant (BG-Lure unit; Biogents AG). Five traps were installed 1.5 metres above ground level, in the same locations where the HLC sampling was performed, and operated for 24 hours.

In SAM, 10 ovitraps baited with an infusion of dried forest floor leaves were installed 1.5 metres above ground level, each containing two wooden paddles. The traps remained in place for 15 days. Wooden paddles bearing mosquito eggs were immersed in water to induce hatching, and the resulting larvae were reared to adult stage in the laboratory for identification.

*Mosquito handling and identification* - After collection, the insects were quickly frozen, transferred to cryogenic tubes, and stored in containers filled with dry ice (-70 ºC). The specimens remained frozen during transport to the laboratory, where they were stored at -80ºC until identification. Mosquitoes were identified under a stereomicroscope on a cold table set at -20ºC, using di-

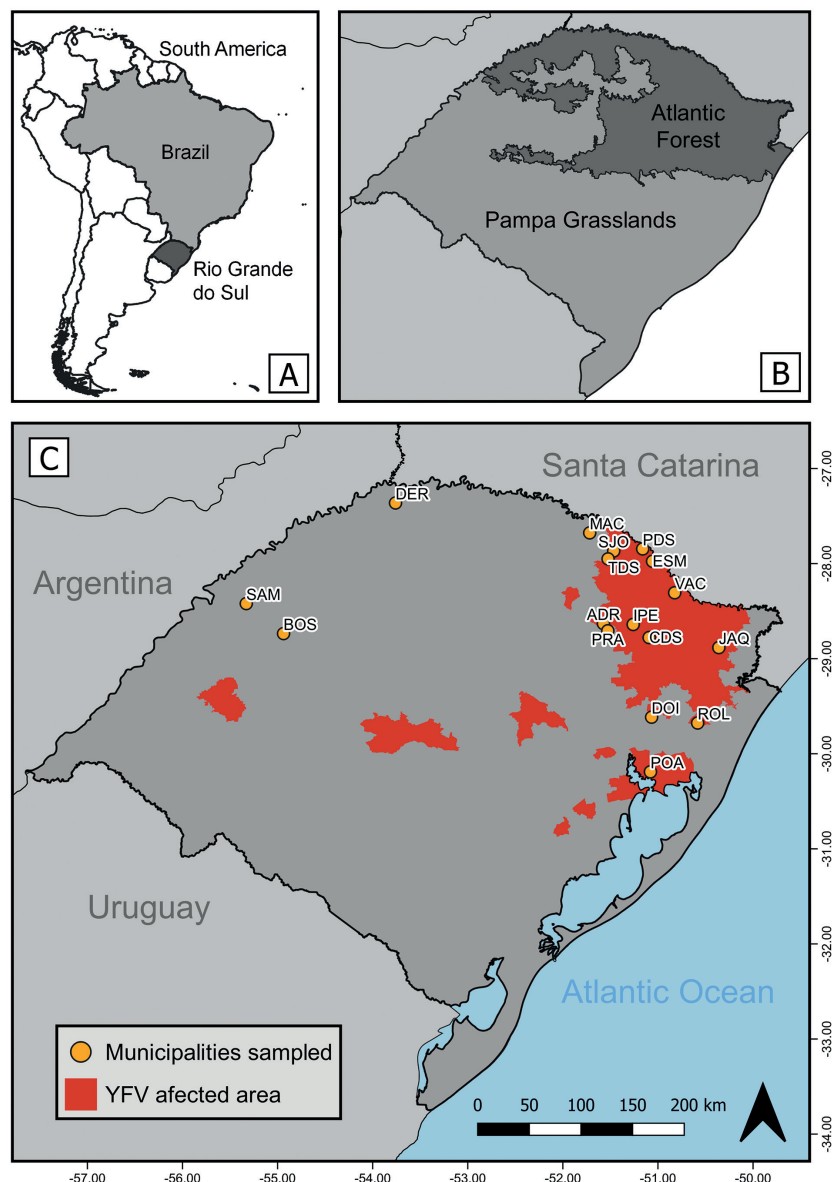

Fig. 1: study area in Rio Grande do Sul (RS) State. (A) Map highlighting the spacial position of Brazil and RS. (B) Biome division of RS (Atlantic Forest and Pampa grasslands). (C) RS map highlighting the affected area by the yellow fever virus (YFV) (in red) according with the criteria established by the Environmental Health Surveillance Division (DVAS) of the Rio Grande do Sul State Health Surveillance Centre (CEVS) and the 17 municipalities where mosquitos are collected (in yellow): André da Rocha (ADR), Bossoroca (BOS), Campestre da Serra (CDS), Derruba-das (DER), Dois Irmãos (DOI), Esmeralda (ESM), Ipê (IPE), Jaquirana (JAQ), Machadinho (MAC), Pinhal da Serra (PDS), Porto Alegre (POA), Protásio Alves (PRA), Rolante (ROL), Santo Antônio das Missões (SAM), São José do Ouro (SJO), Tupanci do Sul (TDS), Vacaria (VAC).

chotomous keys based on female morphological characteristics.[21,22,23] Identified adult specimens were grouped into pools (≤ 10 individuals) according to species, sampling site, and collection date.

*Statistical analysis* - Species richness and abundance indices were calculated for the entire mosquito community based on the total number of captured specimens. In areas where multiple collection methods were applied, the differences in species richness and abundance among methods were evaluated. The average number of specimens captured by each method was assessed for normality using the Shapiro-Wilk test and compared us-

ing analysis of variance (ANOVA), followed by Tukey's post-hoc test. Statistical significance was set at $p < 0.05$. Analyses were performed using R software.

*Virus detection* - Mosquito pools were homogenised in 300 µL of L-15 culture medium containing 1% peni-cillin-streptomycin, and 10% foetal bovine serum (FBS) using the Precellys® 24 tissue homogeniser (Bertin Technologies, France) with glass beads. The homogen-ates were clarified by centrifugation at 12,500 g for 5 min at 4ºC. Supernatants were collected and subjected to RNA extraction using the Extracta® Kit Fast (Loccus, Cotia, Brazil) in the EXTRACTA® 96 automated extrac-

tor (Loccus), following the manufacturer's instructions.

Reverse transcription was performed using the High-Capacity cDNA Reverse Transcription Kit (Applied Biosystems™, Foster City, USA) with 1.5-2 µg of total RNA in a 20 µL reaction, according to the manufacturer's protocol. Quantitative polymerase chain reaction (qPCR) specific for YFV was carried out using primers and probes targeting a 97 bp region of the YFV NS1 gene.[24] To detect other members of the *Orthoflavivirus* genus, an additional qPCR was performed using a pan-*Orthoflavivirus* assay.[25] The list of primers used for amplification is provided in Table I.

To determine the detection limit of the pan-*Orthoflavivirus* assay, serial dilutions of a plasmid (10-10$^7$ copies/µL) were prepared in triplicate. The plasmid was generated by cloning a 260 bp fragment of the NS5 gene (obtained from the YFV vaccine strain) into the TOPO TA vector (Invitrogen), following the manufacturer's instructions. Plasmid concentration was measured by spectrophotometry, and the same plasmid was used as a positive control in all PCR assays. Considering that the first dilution of the control plasmid contained 14 x 10$^8$ genomic copies per microliter of the target fragment, the detection limit obtained was between 1 and 10 copies/reaction. For both assays' amplifications were performed under the following conditions: incubation at 55ºC for 2 min, polymerase activation and initial denaturation at 95ºC for 20 s and 45 cycles of denaturation at 95ºC for 1 s and extension/annealing at 60ºC for 25 s.

## RESULTS

Between February 2021 and February 2022, 17 municipalities in RS were sampled, 14 of which were considered YFV affected areas, based on the detection of the virus in NHPs. In total, 1,210 individuals from the Culicidae family were collected. It was possible to categorize the specimens into 26 taxa; of these, 20 species were identified (Table II).

The most abundant mosquito species was *Psorophora ferox*, followed by *Cx. (Cux.)* spp., comprising 27% and 12% of the mosquitoes collected respectively. These taxa were also the most widespread, being present in more than half of the sampled municipalities (52.9%).

The yellow fever vector *Hg. leucocelaenus* was the fifth most abundant (7%) and was present in almost half of the municipalities sampled (47%). Species of the genus *Sabethes* and *Aedes* also appear among the most collected mosquitoes in this study (Fig. 2).

When comparing the HLC and BG-Pro collection methods, there were no statistical differences between the total mosquito capture averages (p = 0.82). However, when comparing the capture of specific genera, it was observed that the BG-pro traps captured more *Culex*, but these traps did not capture any mosquitoes of the genus *Haemagogus*. When ovitraps were used, 30 individuals were collected, all belonging to the species *Hg. leucocelaenus* (Fig. 3).

In total, 203 pools of mosquitoes, separated by species, location and date of collection, were processed and submitted to a specific RT-qPCR for YFV and to a Pan-*Orthoflavivirus* RT-qPCR. All 203 pools tested negative in both RT-qPCR assays.

## DISCUSSION

At the beginning of 2021, YFV reached RS, the southernmost state in Brazil. Virus spread in different municipalities was accompanied by deaths of native NHPs.[9] Seeking to identify YFV and other arboviruses circulating in sylvatic areas in RS, mosquitoes were obtained from the same sites where circulation of YFV was identified when samples from NHPs were examined.

Regarding the abundance and distribution of the mosquito species found, *Ps. ferox* was the most abundant in the sampled areas. This mosquito is notably aggressive and exhibits a generalist feeding behaviour, obtaining blood from multiple mammalian groups, including NHPs, and to a lesser extent, from birds.[26] Due to this behaviour, this species has been found naturally infected with several viruses, such as YFV, although in vector competence tests it has not been able to transmit this virus.[6] The high proportion of *Ps. ferox* described here corroborates with previous findings in RS.[27] Specimens classified as *Cx. (Cux.)* spp. were the second most abundant and constant in the sampled areas. These were identified to the subgenus level due to the high similarity between species. Members of this group are prob-

## TABLE I
Primers and probes used for yellow fever virus (YFV) and pan-*Orthoflavivirus* real-time quantitative polymerase chain reaction (RT-qPCR) assays

| qPCR assay | Primer & Probe | Sequence (5'- 3') |
|---|---|---|
| YFV | YFallF | GCTAATTGAGGTGYATTGGTCTGC |
| | YFallR | CTGCTAATCGCTCAAMGAACG |
| | YFallP | FAM-ATCGAGTTGCTAGGCAATAAACAC-TAMRA |
| Pan-*Orthoflavivirus* | Flavi all S2 | TACAACATGATGGGMAAACGYGARAA |
| | Flavi all AS4 | GTGTCCCAGCCNGCKGTRTCRTC |
| | Flavi all probe 1 | FAM-TGGTWYATGTGGYTNGGRGC-TAMRA |
| | Flavi all probe 2 | FAM-CCGTGCCATATGGTATATGTGGCTGGGAGC-TAMRA |
| | Flavi all probe 3 | FAM-TTTCTGGAATTTGAAGCCCTGGGTTT-TAMRA |

TABLE II

Abundance of Culicidae collected between February 2021 and January 2022 in 17 municipalities of Rio Grande do Sul

| Taxa | ADR* | BOS | CDS* | DER | DOI* | ESM* | IPE* | JAQ* | MAC | PDS* | POA* | PRA* | ROL* | SAM | SJO* | TDS* | VAC* |
|---|---|---|---|---|---|---|---|---|---|---|---|---|---|---|---|---|---|
| **Anophelinae** | | | | | | | | | | | | | | | | | |
| Anopheles (Anopheles) fluminensis Root, 1927 | | | | 2 | | 2 | | | | | | | | 2 | | | |
| Anopheles (Kerteszia) cruzii Dyar & Knab, 1908 | | | | | | | | | | | 7 | | | | | | |
| Anopheles (Nyssorhynchus) albitarsis Lynch Arribálzaga, 1878 | | | | | | 2 | | | | | | | | | | | |
| Anopheles (Nyssorhynchus) spp. | | | | 1 | | | | | | | | | | | | 3 | |
| **Culicinae** | | | | | | | | | | | | | | | | | |
| **Aedini** | | | | | | | | | | | | | | | | | |
| Aedes (Georgecraigius) fluviatilis (Lutz, 1904) | | | | | 2 | | | | | | 3 | | | | | | |
| Aedes (Ochlerotatus) crinifer (Theobald, 1903) | | | 2 | | | 1 | | 7 | | 18 | | | | | | | |
| Aedes (Ochlerotatus) fulvus (Wiedemann, 1828) | | | | 1 | | | | | | | | | | | | | |
| Aedes (Ochlerotatus) scapularis (Rondani, 1848) | | | | 32 | | | | | | 14 | 10 | | 2 | 1 | | | |
| Aedes (Protomacleaya) terrens (Walker, 1856) | 2 | | | | 2 | | | 1 | | | | | | | | | |
| Aedes (Stegomyia) albopictus (Skuse, 1895) | | | | | 1 | | | | | | 41 | | | | | | |
| Haemagogus (Conopostegus) leucocelaenus (Dyar & Shannon, 1924) | | 13** | | 1 | | 15** | 1** | | | 2** | | 1** | | 51 | | | 1 |
| Psorophora (Janthinosoma) ferox (von Humboldt, 1819) | 2 | 1 | | 138 | | | | 1 | | 92 | 1 | | 31 | 61 | 1 | | 1 |
| **Culicini** | | | | | | | | | | | | | | | | | |
| Culex (Culex) Grupo coronator | | | | | | 2 | | 1 | | | | | | | | 1 | 1 |
| Culex (Culex) spp. | | 4 | | 8 | 1 | 4 | | | | 1 | 1 | | | 118 | | 2 | 2 |
| **Mansoniini** | | | | | | | | | | | | | | | | | |
| Coquillettidia (Rhynchotaenia) spp. | | | | | | | | | | | 5 | | | | | | |
| Coquillettidia (Rhynchotaenia) venezuelensis (Theobald, 1912) | | | | | | 1 | | | | | | | | | | | |
| Mansonia (Mansonia) pseudotitillans (Theobald, 1901) | | | | | | 2 | | | | | | | | | | | |
| Mansonia (Mansonia) titillans (Walker, 1848) | | | | | | | | | | | 97 | | 2 | 1 | | | |
| **Sabethini** | | | | | | | | | | | | | | | | | |
| Sabethes (Peytonulus) aurescens (Lutz, 1905) | | | | | | 126 | 1 | | | | | | | | | | |
| Sabethes (Sabethes) albiprivus Theobald, 1903 | | 7 | | 4 | 1 | 1 | 1 | | | | | 1 | | 56 | | | |
| Sabethes (Sabethes) purpureus (Theobald, 1907) | | | | 1 | | | | | | | | | | 2 | | | |
| Wyeomyia spp. | | 4 | 3 | 9 | | 43 | 2 | | 2 | | | | | | | | |
| Wyeomyia (Phoniomyia) spp. | 2 | | | | | | | | | | | | | | | | |
| Trichoprosopon pallidiventer (Lutz, 1905) | | | 2 | | | 59 | 2 | | 2 | 1 | | | 3 | | | | |
| Wyeomyia (Phoniomyia) davisi (Lane & Cerqueira, 1942) | | | | | | | | 3 | | | | | | | | | |
| Wyeomyia (Phoniomyia) lopesi (Correa & Ramalho, 1956) | | | | | | | | | | | 15 | | | | | | |
| Abundance | 6 | 29 | 7 | 197 | 7 | 258 | 7 | 13 | 4 | 128 | 180 | 2 | 38 | 292 | 1 | 6 | 5 |
| Richness | 3 | 5 | 3 | 10 | 5 | 12 | 5 | 5 | 2 | 6 | 9 | 2 | 4 | 8 | 1 | 3 | 4 |

*Municipalities affected by YFV; **New municipalities with occurence of *Haemagogus leucocelaenus*; municipalities: André da Rocha (ADR), Bossoroca (BOS), Campestre da Serra (CDS), Derrubadas (DER), Dois Irmãos (DOI), Esmeralda (ESM), Ipê (IPE), Jaquirana (JAQ), Machadinho (MAC), Pinhal da Serra (PDS), Porto Alegre (POA), Protásio Alves (PRA), Rolante (ROL), Santo Antônio das Missões (SAM), São José do Ouro (SJO), Tupanci do Sul (TDS), Vacaria (VAC).

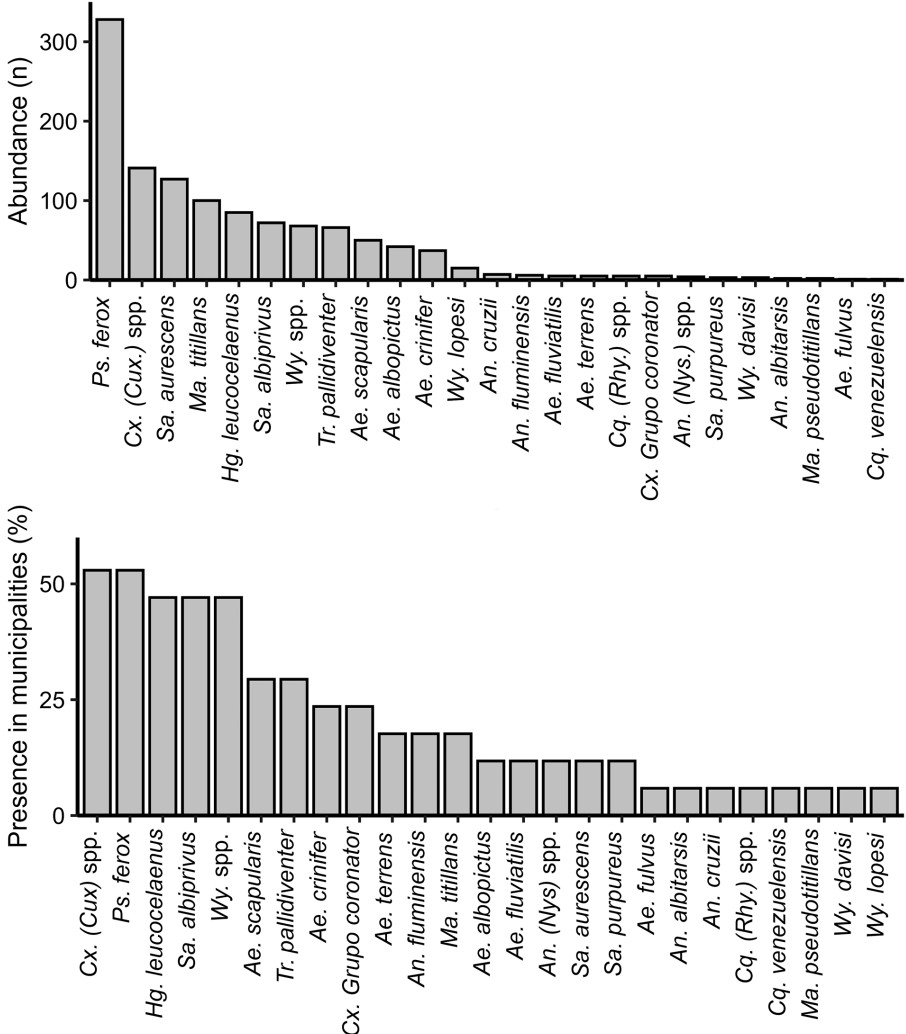

Fig. 2: abundance (A) and presence (B) of 26 mosquito taxa collected between February 2021 and February 2022 in Rio Grande do Sul.

able vectors of SLEV and WNV in South America.[28,29] Corroborating this study, a review shows that this genus is widely distributed throughout RS.[30] *Hg. leucocelaneous* was also frequently found in the sampled areas. This species has a wider distribution, occurring from the Amazon to the southern limit of the Atlantic Forest.[31] It is noteworthy to mention that in Brazil and in other countries the genus *Haemagogus* has a major importance as the main vectors in the YFV sylvatic transmission cycle. The epidemiological importance of *Hg. leucocelaneous* increases in south of Brazil, where other species of this genus are absent.[32] Although not the most abundant, this species has been consistently recorded in studies conducted in the Atlantic Forest of Rio Grande do Sul.[27,32] Our findings reveal the presence of this species in municipalities where it had not been previously recorded, highlighting the potential susceptibility of these areas to YFV circulation. Here, the most abundant and widespread species of genus *Sabethes*, which can also be related to YFV spread, was *Sa. albiprivus*, that has already been found naturally infect-

ed by YFV in Argentina, close to the west border of RS.[33] This species was also found infected during the dry period in the Brazilian Cerrado.[34] Despite limited evidence of natural infections of this species, its role as secondary vector of YFV cannot be neglected. *Sa. purpureus* and *Sa. aurescens* were also found, however, to this date, they have not been found naturally infected by YFV. Members of the genus *Aedes* were also found here, including *Ae. scapularis* and *Ae. albopictus*, both previously found infected with YFV in Brazil.[35,36] The latter is very opportunistic and strongly anthropophilic, so it's frequently captured in urbanised areas and in the transition between forest and city.[37,38] The likelihood of spillover events is closely linked to the ecological and behavioural traits of mosquito species, particularly their host-feeding preferences, population density, and adaptability to anthropogenic environments. For this reason, *Ae. albopictus* is often highlighted as a possible bridge between the urban and sylvatic cycle of YFV.[39] In addition to reports of natural infection, vector competence assays have confirmed its ability to acquire and trans-

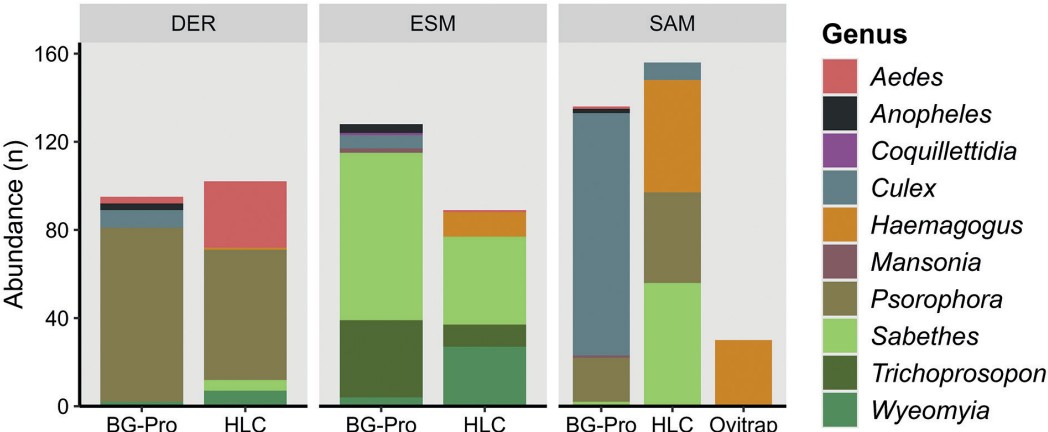

Fig. 3: abundance of different genera collected in the municipalities of Derrubadas (DER), Esmeralda (ESM) and Santo Antônio das Missões (SAM), where collections were carried out with human landing catch (HLC), BG-Pro traps and ovitraps. The bar colours represent the different genera of mosquitoes sampled.

mit YFV, although at low rates.[40] As in previous studies carried out in forested areas of Porto Alegre (POA), the *Ae. albopictus* has been captured in higher abundance than *Hg. leucocelaenus*.[37,38]

Although there were no significant differences between the total average of captures between the BG-Pro traps and HLC, in this study these traps did not capture mosquitoes of the genus *Haemagogus*; however, they captured more *Culex*, which are nocturnal vectors and were captured in smaller quantities manually. These findings are consistent with another study carried out in forest areas in the metropolitan region of RS, where these traps captured a large number of nocturnal vectors, mainly of the *Culex* genus and only one individual of the *Haemagogus* genus.[38] Others studies conducted on the Amazon rainforest also demonstrate low efficiency in capturing *Haemagogus* mosquitoes from BG-Pro traps compared with HCL.[41,42] A study comparing different collection methods carried out in the Atlantic Forest of the State of São Paulo also showed low efficiency in capturing YFV vectors using light-traps baited with the artificial attractant BG-Lure.[43] Even though these traps are highly effective in capturing urban vectors, the few studies using these traps to collect wild mosquitoes in Brazil indicate low efficiency in capturing mosquitoes of the genus *Haemagogus*. The ovitraps installed at SAM captured only *Hg. leucocelanus* eggs, demonstrating the strong potential of this method in monitoring the main vector of YFV in RS. This methodology has already proven effective in capturing *Haemagogus* mosquitoes in the Atlantic Forest.[44] It also enables molecular detection of YFV in mosquito samples after development into adult stage.[45]

In the present study, viral RNA was not detected following application of RT-qPCRs (either when specific primers for YFV or when primers for a wider group of orthoflaviviruses were used). In the zoonotic outbreak of YFV recorded in the RS between 2001 and 2002, the YFV was isolated from the vector *Hg. leucocelaenus*, the infection rate found in that occasion was 8.7%.[10]

The largest YFV outbreak recorded in Rio Grande do Sul occurred between 2008 and 2009. During this period, YFV was again isolated from the vector *Hg. leucocelaenus*, with an infection rate of 3.7%. On that occasion, *Ae. serratus* was also found infected and was considered a potential secondary vector.[32] In the present study, however, this species was not sampled. Notably, these previous studies differed from the current approach by collecting mosquito specimens from a limited number of municipalities and employing intracerebral inoculation in mice as the primary method for viral detection. The low YFV infection rates reported in those studies may explain the absence of YFV-positive samples in the present survey.

Although no positive sample for orthoflaviviruses were detected in the present study, this result agrees with other studies carried out in Brazil. These methods have already been used on mosquito samples collected in urban parks in São Paulo, mainly with *Ae. aegypti* and *Cx.* spp. and, similar to this study, no positive samples were found.[46] These primers and probes were applied to historical samples, the Iguape virus was detected in samples of *Anopheles cruzii*, and Ilheus virus in samples of *Culex* sp., *Coquillettidia juxtamansonia* and *An. triannulatus*.[47] This same RT-qPCR was also used for viral identification after inoculation of mosquito samples from Brazil in cell culture. After inoculation in C6/36 cells it was detected Zika virus in pools of *An. cruzii*, *Limatus durhamii* and *Weomyia confusa*, dengue virus serotype 2 in *Cx.* spp. and *Cx. vaxus* and the insect specific virus (ISV), virus Guapiaçu, from samples of *Ae. scapularis* and *Ae. Terrens*.[48,49]

It is possible that strategies that aim to increase the initial viral load, such as previous inoculation in cell culture or in mice brain, could improve the detection of *Orthoflavivirus* in samples with low viral loads. Furthermore, although orthoflaviviruses were not detected, the possibility of circulation of ISV and arboviruses belonging to other genera, such as *Alphavirus* or *Orthobunyavirus*, cannot be ruled out. In addition, it is important to

highlight that the presence of potential YFV vectors in the study areas indicate the possibility of virus circulation as soon as it reaches enough hosts to establish an outbreak. This fact, by itself, shows how imperative is to keep the surveillance of mosquitoes in areas where previous outbreaks of YFV were recognised.

## ACKNOWLEDGEMENTS

To the contributions of the Divisão de Vigilância Ambiental em Saúde (DVAS/CEVS/SES-RS), which played a central role in preparing for the viral outbreak response and conducting field investigations. We also thank to the Centro de Desenvolvimento Científico e Tecnológico (CDCT/CEVS/SES-RS) for their assistance and access to equipment, and the Laboratório Multiusuário de Biologia Molecular (MultiBioMol/ICBS/UFRGS) for the support provided by their team and for granting access to essential equipment, which greatly contributed to the development of this work.

## AUTHORS' CONTRIBUTION

Conceptualisation - NFDM, JCC, AASC, PMR, FSC, FVSA and ACF; formal analysis - JCC, LVL, MTO, FVSA and ACF; data curation, visualisation, and writing - original draft - NFDM; investigation - NFDM, JCC, AASC, MML, ES, TRM, CHO, ASM, BPG and LVL; methodology - NFDM, JCC, ASM, BPG, LVL, MTO and FVSA; software - NFDM, LVL and MTO; funding acquisition and resources - PMR, FSC and ACF; project administration - PMR, FSC, FVSA and ACF; supervision - JCC and ACF; validation - JCC, MTO, PMR, FSC, FVSA and ACF; writing - review & editing - NFDM, JCC, FVSA and ACF.

## DATA AVAILABILITY

The contents underlying the research text are included in the manuscript.

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

# OPEN PEER REVIEW

Memórias do IOC thanks the anonymous reviewers for their contribution to the peer review of this work.

**FIRST REVIEW ROUND**

REVIEWERS' COMMENTS

### REVIEWER #1

In this manuscript, Müller and colleagues report an entomo-virological investigation of sylvatic yellow fever during an epizootic outbreak in Rio Grande do Sul, Brazil. The study is well-performed and scientifically relevant, especially for the fields of public and environmental health. The quality of the tables and figures is very good. The English is suitable for scientific purposes. However, before publication, I have some suggestions, as detailed below:

- Title: Consider using the term "investigation" instead of "surveillance." Although the authors' work is part of an ongoing surveillance effort, the study reports the investigation of a single outbreak. Just a suggestion.

- Page 4, line 7: Do not begin sentences with acronyms (e.g., YFV). In these cases, the full name should be used.

- Page 8, lines 51 to 60: This paragraph describing sensitivity and other technical aspects of RT-qPCR should be moved to the methods section. Keep in results only the next paragraph ("In total…"), where the results concerning RT-qPCR are indeed described.

- Discussion, page 10, lines 50-55: Please, discuss further the potential impact of opportunistic/anthropophilic mosquitoes on the risk of zoonotic spillover events.

- Supplementary Table I: Enter collection dates in English format.

### REVIEWER #2

In this manuscript, the authors describe entomo-virological surveillance conducted during a sylvatic yellow fever epizootic outbreak, aiming to detect yellow fever virus and other Orthoflavivirus species using molecular tools in mosquitoes from Rio Grande do Sul, Brazil, between 2021 and 2022.

After reading the manuscript, some issues remain unclear.

I suggest including more recent references; for example, citations 1 and 3 appear to be outdated.

Page 3 line 59: The term "incidental hosts" as applied to humans in the context of sylvatic yellow fever requires clarification. Although humans are not part of the natural sylvatic transmission cycle, they are susceptible to infection and can develop disease upon exposure. Moreover, they may become sufficiently viremic to potentially contribute to virus amplification, particularly in certain epidemiological contexts.

Referring to them as incidental may be misleading, as it could suggest a passive or irrelevant role, which is not accurate. If the authors intend to emphasize that human infections occur sporadically or do not sustain the sylvatic cycle, it would be helpful to rephrase this for clarity, to avoid the impression that humans play no role in transmission dynamics.

Pag 4 line 9: "…from 2014":  There seems to be a discrepancy regarding the start of the YFV outbreak in Brazil. The text mentions 2014, but most sources report it began in 2016. Could the authors clarify this?

Page 4 line 27:  . In the Introduction, it may be important to include a paragraph summarizing the epidemiological situation in Rio Grande do Sul during 2019–2021 ( or 2022). This could include the number of confirmed human cases, the number of epizootics detected in the region, and the observed impact on non-human primate (NHP) mortality. Providing this background would help contextualize the study and clearly define its objectives.

Page 4, lines 37–53: The authors present evidence of the circulation of other Orthoflavivirus species; however, it would be helpful—if available—to include epidemiological data on human cases during the 2021–2022 period. This information would contribute to a clearer definition of the study's objective and help contextualize the virological findings within the broader public health framework.

Page 5 line 30. It should be NHP

Page 5 line 30.  In order to clarify the objective, it would be helpful to provide information related to the epizootics that occurred during 2021–2022, when the entomological studies were carried out. The introduction currently lacks epidemiological information regarding the epizootic events during the period in which the mosquito captures took place.

Page 5, lines 40–54: What was the duration of the manual mosquito capture efforts at each affected site?

Page 5 line 57- page 6 line 21: Why were mosquito collections not conducted uniformly across all municipalities? Providing an explanation for the differences in sampling effort or methodology between locations would help clarify the study design and support the interpretation of the results (figure 3).

Page 7 line 11.  Centrifugal force should be expressed in g (relative centrifugal force, RCF), as this allows for standardization regardless of rotor diameter.

Page 8, lines 41–48: Was the comparison made during the same time periods? Perhaps the BG-Pro traps did not capture any Haemagogus mosquitoes due to their placement. Is it possible that these traps were located inside houses or in other suboptimal sites? What might have been the results if the authors had placed the traps in the

canopy of trees, in the same locations where the ovitraps were installed?

Page 9, lines 17-19: Once again, the Introduction lacks sufficient epidemiological information regarding the outbreak that affected Rio Grande do Sul. Including detailed data on the extent and impact of the outbreak would strengthen the context and justification of the study.

Page 10 line 50: In the phrase "…the transition between forest and city.(31,32 For this reason,…" the parentheses around the citations are not properly closed. Please close the parentheses for references (31, 32).

Page 11 line 10- 17: I suggest that the conclusions presented here be reconsidered, given that the sampling effort and trap placement were not uniform across all locations.

Page 12 line 20-30: Did the authors perform mice inoculation in order to isolate viruses? Clarification on this point would be helpful.

Figure 2: Were the samples collected from February 2021 to February 2022 or until January 2022? Clarification on this timeline would be helpful. In addition, it is difficult to understand part B. Is the percentage shown for all cities? What did the authors intend to explain with this figure?

Figure 3: Why does the comparison of different sampling methods include only three cities? Were the same methods not applied across all sampled locations? Providing clarification on the sampling design would improve understanding of the results.

Supplementary Table I: The table presents information only for the year 2021. Is there missing data for 2022, or was data for that year not collected? Clarification on this point would be helpful.

## AUTHORS' RESPONSE TO THE REVIEWERS

We sincerely thank the reviewers for their valuable comments and insightful suggestions. Their feedback has greatly contributed to improving the quality and clarity of our manuscript. We carefully considered each point raised by the reviewers and implemented several modifications were made that we believe have strengthened the work.

Kind regards,

The research team

REVIEWER COMMENTS:

Reviewer 1:

In this manuscript, Müller and colleagues report an entomo-virological investigation of sylvatic yellow fever during an epizootic outbreak in Rio Grande do Sul, Brazil. The study is well-performed and scientifically relevant, especially for the fields of public and environmental health. The quality of the tables and figures is very good. The English is suitable for scientific purposes. However, before publication, I have some suggestions, as detailed below:

- Title: Consider using the term "investigation" instead of "surveillance." Although the authors' work is part of an ongoing surveillance effort, the study reports the investigation of a single outbreak. Just a suggestion.

The term "surveillance" has been replaced with "investigation" throughout the manuscript to more accurately reflect the scope of the study.

- Page 4, line 7: Do not begin sentences with acronyms (e.g., YFV). In these cases, the full name should be used.

The manuscript has been revised to ensure that sentences do not begin with acronyms.

- Page 8, lines 51 to 60: This paragraph describing sensitivity and other technical aspects of RT-qPCR should be moved to the methods section. Keep in results only the next paragraph ("In total…"), where the results concerning RT-qPCR are indeed described.

The paragraph describing the sensitivity and technical aspects of RT-qPCR has been relocated to the Methods section.

- Discussion, page 10, lines 50-55: Please, discuss further the potential impact of opportunistic/anthropophilic mosquitoes on the risk of zoonotic spillover events.

To address this point, we have expanded the discussion by adding the following sentence:

"The likelihood of spillover events is closely linked to the ecological and behavioral traits of mosquito species, particularly their host-feeding preferences, population density, and adaptability to anthropogenic environments."

- Supplementary Table I: Enter collection dates in English format.

The collection dates in Supplementary Table I have been revised to follow the English date format.

Reviewer 2:

In this manuscript, the authors describe entomo-virological surveillance conducted during a sylvatic yellow fever epizootic outbreak, aiming to detect yellow fever virus and other Orthoflavivirus species using molecular tools in mosquitoes from Rio Grande do Sul, Brazil, between 2021 and 2022.

After reading the manuscript, some issues remain unclear.

I suggest including more recent references; for example, citations 1 and 3 appear to be outdated.

A portion of the literature has been updated to reflect recent findings. However, certain references, were retained as they represent foundational and original contributions to the field. These works provide essential context and accurately describe the epidemiological conditions at the time they were produced, which remain relevant to the historical and scientific framing of our study.

Page 3 line 59: The term "incidental hosts" as applied to humans in the context of sylvatic yellow fever requires

clarification. Although humans are not part of the natural sylvatic transmission cycle, they are susceptible to infection and can develop disease upon exposure. Moreover, they may become sufficiently viremic to potentially contribute to virus amplification, particularly in certain epidemiological contexts. Referring to them as incidental may be misleading, as it could suggest a passive or irrelevant role, which is not accurate. If the authors intend to emphasize that human infections occur sporadically or do not sustain the sylvatic cycle, it would be helpful to rephrase this for clarity, to avoid the impression that humans play no role in transmission dynamics.

The terminology has been revised to improve clarity and to better reflect the role of humans in the sylvatic yellow fever cycle. The revised text now states: *"Humans, within the sylvatic cycle, are considered incidental hosts, with infections occurring sporadically, developing disease and viremia, although they generally do not sustain transmission."* This modification acknowledges the potential for human involvement in virus amplification while emphasizing their non-sustaining role in the sylvatic transmission cycle.

Pag 4 line 9: "…from 2014": There seems to be a discrepancy regarding the start of the YFV outbreak in Brazil. The text mentions 2014, but most sources report it began in 2016. Could the authors clarify this?

*The reference to 2014 was intended to highlight the initial detection of YFV expansion beyond the endemic Amazon region, which occurred in less populated areas and resulted in limited human cases and NPH deaths detection. However, the widespread outbreak involving both humans and NHPs began in 2016, when the virus reached more densely populated areas, leading to increased media coverage and scientific output. Minor revisions have been made to the paragraph to improve clarity.*

Page 4 line 27: In the Introduction, it may be important to include a paragraph summarizing the epidemiological situation in Rio Grande do Sul during 2019–2021 (or 2022). This could include the number of confirmed human cases, the number of epizootics detected in the region, and the observed impact on non-human primate (NHP) mortality. Providing this background would help contextualize the study and clearly define its objectives.

In accordance, we have added a paragraph to the Introduction summarizing the epidemiological situation of YFV in RS. This includes historical outbreak data, the detection of epizootics among NHPs, and the human cases.

Page 4, lines 37–53: The authors present evidence of the circulation of other Orthoflavivirus species; however, it would be helpful—if available—to include epidemiological data on human cases during the 2021–2022 period. This information would contribute to a clearer definition of the study's objective and help contextualize the virological findings within the broader public health framework.

We have incorporated epidemiological data on human cases of *Orthoflavivirus* infections during the 2021–2022 period. Specifically, we added information regarding the occurrence of urban arboviruses (dengue and Zika) reported in human populations during the study period. Furthermore, the original paragraph describing the detection of *Orthoflavivirus* species in wild and livestock animals was retained, with a clarification that these findings represent historical evidence.

Page 5 line 30. It should be NHP

The revised acronym now states: "NHP".

Page 5 line 30. In order to clarify the objective, it would be helpful to provide information related to the epizootics that occurred during 2021–2022, when the entomological studies were carried out. The introduction currently lacks epidemiological information regarding the epizootic events during the period in which the mosquito captures took place.

As noted in our previous response, we have incorporated detailed information regarding the YFV outbreak in RS into the Introduction.

Page 5, lines 40–54: What was the duration of the manual mosquito capture efforts at each affected site?

We have updated the supplementary table to include the specific duration of mosquito collection at each location.

Page 5 line 57- page 6 line 21: Why were mosquito collections not conducted uniformly across all municipalities? Providing an explanation for the differences in sampling effort or methodology between locations would help clarify the study design and support the interpretation of the results (figure 3).

Mosquito collections were not standardized across all municipalities due to the emergency nature of the field activities, which were conducted during an active YFV outbreak as part of epizootic investigation procedures. The field team adapted their sampling strategy in response to real-time notifications of sick or dead NHP, relocating to newly affected forest areas as needed. In several instances, primate samples were collected directly by municipal teams, and vector sampling could not be performed at those sites. Additionally, mosquito collections were extended to neighboring municipalities that had not reported epizootics but were ecologically connected to affected areas, as well as to municipalities with confirmed YFV circulation in previous years. To enhance transparency, the supplementary table now includes the duration of mosquito collection at each sampling site, and the manuscript text has been revised to clarify these methodological considerations.

Page 7 line 11. Centrifugal force should be expressed in g (relative centrifugal force, RCF), as this allows for standardization regardless of rotor diameter.

The revised version now states: "12,500 g".

Page 8, lines 41–48: Was the comparison made during the same time periods? Perhaps the BG-Pro traps did not capture any Haemagogus mosquitoes due to their placement. Is it possible that these traps were located inside houses or in other suboptimal sites? What might have been the results if the authors had placed the traps in the canopy of trees, in the same locations where the ovitraps were installed?

When more than one sampling method was employed, all were applied within the same forest fragment. The methodology section has been revised to clarify this aspect.

Page 9, lines 17-19: Once again, the Introduction lacks sufficient epidemiological information regarding the outbreak that affected Rio Grande do Sul. Including detailed data on the extent and impact of the outbreak would strengthen the context and justification of the study.

As noted in our previous response, we have incorporated detailed information regarding the YFV outbreak in RS into the Introduction.

Page 10 line 50: In the phrase "…the transition between forest and city.(31,32 For this reason,…" the parentheses around the citations are not properly closed. Please close the parentheses for references (31, 32).

The parentheses around references "(31, 32)" have been properly closed.

Page 11 line 10-17: I suggest that the conclusions presented here be reconsidered, given that the sampling effort and trap placement were not uniform across all locations.

We respectfully acknowledge the reviewer's suggestion. However, we have chosen to maintain the conclusions as originally presented. Collections were conducted within the same forest fragments, ensuring consistency in environmental context. Although the primary objective of this study was not to evaluate collection methodologies, the findings obtained during our investigation, alongside supporting evidence from the literature, are relevant for clarifying the limitations of BG-Pro traps in capturing sylvatic yellow fever vectors. While BG-Pro traps are highly effective for urban vector surveillance, our results reinforce that they do not perform adequately for *Haemagogus* spp. under the conditions tested. We believe this observation contributes meaningfully to the understanding of YFV vector surveillance strategies in forested environments.

Page 12 line 20-30: Did the authors perform mice inoculation in order to isolate viruses? Clarification on this point would be helpful.

Mice inoculation was not performed in the present study. The manuscript has been revised to explicitly clarify this aspect. Our virological analyses were limited to RT-qPCR detection.

Figure 2: Were the samples collected from February 2021 to February 2022 or until January 2022? Clarification on this timeline would be helpful. In addition, it is difficult to understand part B. Is the percentage shown for all cities? What did the authors intend to explain with this figure?

The samples were collected from February 2021 through February 2022. The figure legend has been revised to correct the previously unclear wording regarding the sampling period. Regarding Figure 2B, the percentages represent the presence of each vector species across the total sampled municipalities. A higher percentage indicates that a given species was detected in a greater number of municipalities. The primary objective of this figure is to illustrate the relative dissemination of each vector species throughout the sampled territory of RS during the study period.

Figure 3: Why does the comparison of different sampling methods include only three cities? Were the same methods not applied across all sampled locations? Providing clarification on the sampling design would improve understanding of the results.

The comparison presented in Figure 3 includes only three municipalities because the only sampling method applied across all locations was human landing catch (HLC). Other methods were used selectively, depending on logistical feasibility and local conditions during the outbreak response. This rationale is now clarified in the methodology section following recent revisions, and we believe the current version of the manuscript adequately reflects the sampling design and its implications for data interpretation.

Supplementary Table I: The table presents information only for the year 2021. Is there missing data for 2022, or was data for that year not collected? Clarification on this point would be helpful.

Data collection was primarily conducted in 2021, during the peak of the yellow fever outbreak. However, some areas (PDS and ESM) were revisited approximately one year after the initial epizootic events. These follow-up collections are indicated in the supplementary table. No additional data collection campaigns were conducted in 2022 beyond these targeted revisits.

## SECOND REVIEW ROUND

REVIEWERS' COMMENTS

### REVIEWER #1

The authors answered to my comments satisfactorily.

### REVIEWER #2

No comments.

