## [Reviewer Report · FIRST REVIEW ROUND - REVIEWERS COMMENTS]

## REVIEWER #1

In this manuscript, Müller and colleagues report an entomo-virological investigation of sylvatic yellow fever during an epizootic outbreak in Rio Grande do Sul, Brazil. The study is well-performed and scientifically relevant, especially for the fields of public and environmental health. The quality of the tables and figures is very good. The English is suitable for scientific purposes. However, before publication, I have some suggestions, as detailed below:

- Title: Consider using the term "investigation" instead of "surveillance." Although the authors' work is part of an ongoing surveillance effort, the study reports the investigation of a single outbreak. Just a suggestion.

- Page 4, line 7: Do not begin sentences with acronyms (e.g., YFV). In these cases, the full name should be used.

- Page 8, lines 51 to 60: This paragraph describing sensitivity and other technical aspects of RT-qPCR should be moved to the methods section. Keep in results only the next paragraph ("In total…"), where the results concerning RT-qPCR are indeed described.

- Discussion, page 10, lines 50-55: Please, discuss further the potential impact of opportunistic/anthropophilic mosquitoes on the risk of zoonotic spillover events.

- Supplementary Table I: Enter collection dates in English format.

## REVIEWER #2

In this manuscript, the authors describe entomo-virological surveillance conducted during a sylvatic yellow fever epizootic outbreak, aiming to detect yellow fever virus and other Orthoflavivirus species using molecular tools in mosquitoes from Rio Grande do Sul, Brazil, between 2021 and 2022.

After reading the manuscript, some issues remain unclear.

I suggest including more recent references; for example, citations 1 and 3 appear to be outdated.

Page 3 line 59: The term "incidental hosts" as applied to humans in the context of sylvatic yellow fever requires clarification. Although humans are not part of the natural sylvatic transmission cycle, they are susceptible to infection and can develop disease upon exposure. Moreover, they may become sufficiently viremic to potentially contribute to virus amplification, particularly in certain epidemiological contexts. Referring to them as incidental may be misleading, as it could suggest a passive or irrelevant role, which is not accurate. If the authors intend to emphasize that human infections occur sporadically or do not sustain the sylvatic cycle, it would be helpful to rephrase this for clarity, to avoid the impression that humans play no role in transmission dynamics.

Pag 4 line 9: "…from 2014": There seems to be a discrepancy regarding the start of the YFV outbreak in Brazil. The text mentions 2014, but most sources report it began in 2016. Could the authors clarify this?

Page 4 line 27: In the Introduction, it may be important to include a paragraph summarizing the epidemiological situation in Rio Grande do Sul during 2019–2021 (or 2022). This could include the number of confirmed human cases, the number of epizootics detected in the region, and the observed impact on non-human primate (NHP) mortality. Providing this background would help contextualize the study and clearly define its objectives.

Page 4, lines 37–53: The authors present evidence of the circulation of other Orthoflavivirus species; however, it would be helpful—if available—to include epidemiological data on human cases during the 2021–2022 period. This information would contribute to a clearer definition of the study's objective and help contextualize the virological findings within the broader public health framework.

Page 5 line 30. It should be NHP

Page 5 line 30. In order to clarify the objective, it would be helpful to provide information related to the epizootics that occurred during 2021–2022, when the entomological studies were carried out. The introduction currently lacks epidemiological information regarding the epizootic events during the period in which the mosquito captures took place.

Page 5, lines 40–54: What was the duration of the manual mosquito capture efforts at each affected site?

Page 5 line 57- page 6 line 21: Why were mosquito collections not conducted uniformly across all municipalities? Providing an explanation for the differences in sampling effort or methodology between locations would help clarify the study design and support the interpretation of the results (figure 3).

Page 7 line 11. Centrifugal force should be expressed in g (relative centrifugal force, RCF), as this allows for standardization regardless of rotor diameter.

Page 8, lines 41–48: Was the comparison made during the same time periods? Perhaps the BG-Pro traps did not capture any Haemagogus mosquitoes due to their placement. Is it possible that these traps were located inside houses or in other suboptimal sites? What might have been the results if the authors had placed the traps in the canopy of trees, in the same locations where the ovitraps were installed?

Page 9, lines 17-19: Once again, the Introduction lacks sufficient epidemiological information regarding the outbreak that affected Rio Grande do Sul. Including detailed data on the extent and impact of the outbreak would strengthen the context and justification of the study.

Page 10 line 50: In the phrase "…the transition between forest and city.(31,32 For this reason,…" the parentheses around the citations are not properly closed. Please close the parentheses for references (31, 32).

Page 11 line 10-17: I suggest that the conclusions presented here be reconsidered, given that the sampling effort and trap placement were not uniform across all locations.

Page 12 line 20-30: Did the authors perform mice inoculation in order to isolate viruses? Clarification on this point would be helpful.

Figure 2: Were the samples collected from February 2021 to February 2022 or until January 2022? Clarification on this timeline would be helpful. In addition, it is difficult to understand part B. Is the percentage shown for all cities? What did the authors intend to explain with this figure?

Figure 3: Why does the comparison of different sampling methods include only three cities? Were the same methods not applied across all sampled locations? Providing clarification on the sampling design would improve understanding of the results.

Supplementary Table I: The table presents information only for the year 2021. Is there missing data for 2022, or was data for that year not collected? Clarification on this point would be helpful.

## AUTHORS' RESPONSE TO THE REVIEWERS

We sincerely thank the reviewers for their valuable comments and insightful suggestions. Their feedback has greatly contributed to improving the quality and clarity of our manuscript. We carefully considered each point raised by the reviewers and implemented several modifications were made that we believe have strengthened the work.

Kind regards,

The research team

## REVIEWER COMMENTS:

Reviewer 1:

In this manuscript, Müller and colleagues report an entomo-virological investigation of sylvatic yellow fever during an epizootic outbreak in Rio Grande do Sul, Brazil. The study is well-performed and scientifically relevant, especially for the fields of public and environmental health. The quality of the tables and figures is very good. The English is suitable for scientific purposes. However, before publication, I have some suggestions, as detailed below:

*- Title: Consider using the term "investigation" instead of "surveillance." Although the authors' work is part of an ongoing surveillance effort, the study reports the investigation of a single outbreak. Just a suggestion.*

The term "surveillance" has been replaced with "investigation" throughout the manuscript to more accurately reflect the scope of the study.

*- Page 4, line 7: Do not begin sentences with acronyms (e.g., YFV). In these cases, the full name should be used.*

The manuscript has been revised to ensure that sentences do not begin with acronyms.

*- Page 8, lines 51 to 60: This paragraph describing sensitivity and other technical aspects of RT-qPCR should be moved to the methods section. Keep in results only the next paragraph ("In total…"), where the results concerning RT-qPCR are indeed described.*

The paragraph describing the sensitivity and technical aspects of RT-qPCR has been relocated to the Methods section.

*- Discussion, page 10, lines 50-55: Please, discuss further the potential impact of opportunistic/anthropophilic mosquitoes on the risk of zoonotic spillover events.*

To address this point, we have expanded the discussion by adding the following sentence:

"The likelihood of spillover events is closely linked to the ecological and behavioral traits of mosquito species, particularly their host-feeding preferences, population density, and adaptability to anthropogenic environments."

*- Supplementary Table I: Enter collection dates in English format.*

The collection dates in Supplementary Table I have been revised to follow the English date format.

## Reviewer 2:

In this manuscript, the authors describe entomo-virological surveillance conducted during a sylvatic yellow fever epizootic outbreak, aiming to detect yellow fever virus and other Orthoflavivirus species using molecular tools in mosquitoes from Rio Grande do Sul, Brazil, between 2021 and 2022.

After reading the manuscript, some issues remain unclear.

*I suggest including more recent references; for example, citations 1 and 3 appear to be outdated.*

A portion of the literature has been updated to reflect recent findings. However, certain references, were retained as they represent foundational and original contributions to the field. These works provide essential context and accurately describe the epidemiological conditions at the time they were produced, which remain relevant to the historical and scientific framing of our study.

*Page 3 line 59: The term "incidental hosts" as applied to humans in the context of sylvatic yellow fever requires clarification. Although humans are not part of the natural sylvatic transmission cycle, they are susceptible to infection and can develop disease upon exposure. Moreover, they may become sufficiently viremic to potentially contribute to virus amplification, particularly in certain epidemiological contexts. Referring to them as incidental may be misleading, as it could suggest a passive or irrelevant role, which is not accurate. If the authors intend to emphasize that human infections occur sporadically or do not sustain the sylvatic cycle, it would be helpful to rephrase this for clarity, to avoid the impression that humans play no role in transmission dynamics.*

The terminology has been revised to improve clarity and to better reflect the role of humans in the sylvatic yellow fever cycle. The revised text now states: "Humans, within the sylvatic cycle, are considered incidental hosts, with infections occurring sporadically, developing disease and viremia, although they generally do not sustain transmission." This modification acknowledges the potential for human involvement in virus amplification while emphasizing their non-sustaining role in the sylvatic transmission cycle.

*Pag 4 line 9: "…from 2014": There seems to be a discrepancy regarding the start of the YFV outbreak in Brazil. The text mentions 2014, but most sources report it began in 2016. Could the authors clarify this?*

The reference to 2014 was intended to highlight the initial detection of YFV expansion beyond the endemic Amazon region, which occurred in less populated areas and resulted in limited human cases and NPH deaths detection. However, the widespread outbreak involving both humans and NHPs began in 2016, when the virus reached more densely populated areas, leading to increased media coverage and scientific output. Minor revisions have been made to the paragraph to improve clarity.

*Page 4 line 27: In the Introduction, it may be important to include a paragraph summarizing the epidemiological situation in Rio Grande do Sul during 2019–2021 (or 2022). This could include the number of confirmed human cases, the number of epizootics detected in the region, and the observed impact on non-human primate (NHP) mortality. Providing this background would help contextualize the study and clearly define its objectives.*

In accordance, we have added a paragraph to the Introduction summarizing the epidemiological situation of YFV in RS. This includes historical outbreak data, the detection of epizootics among NHPs, and the human cases.

*Page 4, lines 37–53: The authors present evidence of the circulation of other Orthoflavivirus species; however, it would be helpful—if available—to include epidemiological data on human cases during the 2021–2022 period. This information would contribute to a clearer definition of the study's objective and help contextualize the virological findings within the broader public health framework.*

We have incorporated epidemiological data on human cases of Orthoflavivirus infections during the 2021–2022 period. Specifically, we added information regarding the occurrence of urban arboviruses (dengue and Zika) reported in human populations during the study period. Furthermore, the original paragraph describing the detection of Orthoflavivirus species in wild and livestock animals was retained, with a clarification that these findings represent historical evidence.

*Page 5 line 30. It should be NHP*

The revised acronym now states: "NHP".

*Page 5 line 30. In order to clarify the objective, it would be helpful to provide information related to the epizootics that occurred during 2021–2022, when the entomological studies were carried out. The introduction currently lacks epidemiological information regarding the epizootic events during the period in which the mosquito captures took place.*

As noted in our previous response, we have incorporated detailed information regarding the YFV outbreak in RS into the Introduction.

*Page 5, lines 40–54: What was the duration of the manual mosquito capture efforts at each affected site?*

We have updated the supplementary table to include the specific duration of mosquito collection at each location.

*Page 5 line 57- page 6 line 21: Why were mosquito collections not conducted uniformly across all municipalities? Providing an explanation for the differences in sampling effort or methodology between locations would help clarify the study design and support the interpretation of the results (figure 3).*

Mosquito collections were not standardized across all municipalities due to the emergency nature of the field activities, which were conducted during an active YFV outbreak as part of epizootic investigation procedures. The field team adapted their sampling strategy in response to real-time notifications of sick or dead NHP, relocating to newly affected forest areas as needed. In several instances, primate samples were collected directly by municipal teams, and vector sampling could not be performed at those sites. Additionally, mosquito collections were extended to neighboring municipalities that had not reported epizootics but were ecologically connected to affected areas, as well as to municipalities with confirmed YFV circulation in previous years. To enhance transparency, the supplementary table now includes the duration of mosquito collection at each sampling site, and the manuscript text has been revised to clarify these methodological considerations.

*Page 7 line 11. Centrifugal force should be expressed in g (relative centrifugal force, RCF), as this allows for standardization regardless of rotor diameter.*

The revised version now states: "12,500 g".

*Page 8, lines 41–48: Was the comparison made during the same time periods? Perhaps the BG-Pro traps did not capture any Haemagogus mosquitoes due to their placement. Is it possible that these traps were located inside houses or in other suboptimal sites? What might have been the results if the authors had placed the traps in the canopy of trees, in the same locations where the ovitraps were installed?*

When more than one sampling method was employed, all were applied within the same forest fragment. The methodology section has been revised to clarify this aspect.

*Page 9, lines 17-19: Once again, the Introduction lacks sufficient epidemiological information regarding the outbreak that affected Rio Grande do Sul. Including detailed data on the extent and impact of the outbreak would strengthen the context and justification of the study.*

As noted in our previous response, we have incorporated detailed information regarding the YFV outbreak in RS into the Introduction.

*Page 10 line 50: In the phrase "…the transition between forest and city.(31,32 For this reason,…" the parentheses around the citations are not properly closed. Please close the parentheses for references (31, 32).*

The parentheses around references "(31, 32)" have been properly closed.

*Page 11 line 10-17: I suggest that the conclusions presented here be reconsidered, given that the sampling effort and trap placement were not uniform across all locations.*

We respectfully acknowledge the reviewer's suggestion. However, we have chosen to maintain the conclusions as originally presented. Collections were conducted within the same forest fragments, ensuring consistency in environmental context. Although the primary objective of this study was not to evaluate collection methodologies, the findings obtained during our investigation, alongside supporting evidence from the literature, are relevant for clarifying the limitations of BG-Pro traps in capturing sylvatic yellow fever vectors. While BG-Pro traps are highly effective for urban vector surveillance, our results reinforce that they do not perform adequately for Haemagogus spp. under the conditions tested. We believe this observation contributes meaningfully to the understanding of YFV vector surveillance strategies in forested environments.

*Page 12 line 20-30: Did the authors perform mice inoculation in order to isolate viruses? Clarification on this point would be helpful.*

Mice inoculation was not performed in the present study. The manuscript has been revised to explicitly clarify this aspect. Our virological analyses were limited to RT-qPCR detection.

*Figure 2: Were the samples collected from February 2021 to February 2022 or until January 2022? Clarification on this timeline would be helpful. In addition, it is difficult to understand part B. Is the percentage shown for all cities? What did the authors intend to explain with this figure?*

The samples were collected from February 2021 through February 2022. The figure legend has been revised to correct the previously unclear wording regarding the sampling period. Regarding Figure 2B, the percentages represent the presence of each vector species across the total sampled municipalities. A higher percentage indicates that a given species was detected in a greater number of municipalities. The primary objective of this figure is to illustrate the relative dissemination of each vector species throughout the sampled territory of RS during the study period.

*Figure 3: Why does the comparison of different sampling methods include only three cities? Were the same methods not applied across all sampled locations? Providing clarification on the sampling design would improve understanding of the results.*

The comparison presented in Figure 3 includes only three municipalities because the only sampling method applied across all locations was human landing catch (HLC). Other methods were used selectively, depending on logistical feasibility and local conditions during the outbreak response. This rationale is now clarified in the methodology section following recent revisions, and we believe the current version of the manuscript adequately reflects the sampling design and its implications for data interpretation.

*Supplementary Table I: The table presents information only for the year 2021. Is there missing data for 2022, or was data for that year not collected? Clarification on this point would be helpful.*

Data collection was primarily conducted in 2021, during the peak of the yellow fever outbreak. However, some areas (PDS and ESM) were revisited approximately one year after the initial epizootic events. These follow-up collections are indicated in the supplementary table. No additional data collection campaigns were conducted in 2022 beyond these targeted revisits.